# Enhancing Insights into Australia’s Gonococcal Surveillance Programme through Stochastic Modelling

**DOI:** 10.3390/pathogens12070907

**Published:** 2023-07-04

**Authors:** Phu Cong Do, Yibeltal Assefa Alemu, Simon Andrew Reid

**Affiliations:** School of Public Health, Faculty of Medicine, University of Queensland, Herston, QLD 4006, Australia

**Keywords:** antimicrobial resistance, surveillance, *Neisseria gonorrhoeae*, scenario tree modelling, stochastic modelling

## Abstract

Continued surveillance of antimicrobial resistance is critical as a feedback mechanism for the generation of concerted public health action. A characteristic of importance in evaluating disease surveillance systems is representativeness. Scenario tree modelling offers an approach to quantify system representativeness. This paper utilises the modelling approach to assess the Australian Gonococcal Surveillance Programme’s representativeness as a case study. The model was built by identifying the sequence of events necessary for surveillance output generation through expert consultation and literature review. A scenario tree model was developed encompassing 16 dichotomous branches representing individual system sub-components. Key classifications included biological sex, clinical symptom status, and location of healthcare service access. The expected sensitivities for gonococcal detection and antibiotic status ascertainment were 0.624 (95% CI; 0.524, 0.736) and 0.144 (95% CI; 0.106, 0.189), respectively. Detection capacity of the system was observed to be high overall. The stochastic modelling approach has highlighted the need to consider differential risk factors such as sex, health-seeking behaviours, and clinical behaviour in sample generation. Actionable points generated by this study include modification of clinician behaviour and supplementary systems to achieve a greater contextual understanding of the surveillance data generation process.

## 1. Introduction

Antimicrobial resistance (AMR) is a growing public health threat that poses a risk to the use of antimicrobials [1]. Early estimates project mortality due to AMR to reach 10 million per year by 2050 in the absence of effective action [2]. The global threat posed by AMR has prompted concerted efforts by the World Health Organization (WHO) to implement a global action plan [3]. This initiative has provided the impetus for member states of the WHO to develop context-specific plans to ensure the viability of therapeutics. A critical element of those plans is the necessity of surveillance to understand the epidemiology of AMR, monitor the effectiveness of interventions, and to identify emerging threats [3].

Surveillance is a fundamental component of communicable disease control [4]. The role of surveillance is critical in the generation and provision of information that informs any public health action [5]. The exigencies of concerted endeavours require the quality of surveillance data to be reliable to support greater comprehensiveness of the situation. In the context of AMR, surveillance is an essential feedback mechanism for governance [3]. Despite the wide acceptance of surveillance as a key component in AMR mitigation efforts, there exists considerable variability across national contexts regarding surveillance system structures [6,7,8]. As a result, the comparability of AMR data is limited by its representativeness. Evaluative frameworks for public health surveillance, such as that of the Centres for Disease Control and Prevention (CDC), have recognised the importance of representativeness but facilitate evaluation from a subjective perspective [9]. The absence of a standardised and objective methodology for assessing AMR surveillance system representativeness will continue to impede the comparability of data and states of AMR across global contexts.

Scenario tree modelling (STM) is an established technique that has been utilised in the decision analysis and risk assessment spaces to understand the outcomes following sequential events throughout a system by composing a series of trees defined by positive and negative outcomes [10,11]. In the context of AMR surveillance systems, the adoption of the STM has yet to be demonstrated. However, there is unrealised potential in its application to further refine AMR surveillance system representativeness. In the wider literature, STM has been used to substantiate freedom from disease by examining the surveillance data generation process and identifying the potential for the disease to not be captured [11]. The proposed benefit of STM is that the methodology necessitates the systematic deconstruction of the data generation process to understand the sequence of events that are implicated [10]. With the understanding of how surveillance data are generated, discussion regarding the representativeness of the system can focus on identifying leverage points to improve. At the time of publication of this article, there has yet to be any literature which has applied this methodology to AMR surveillance.

The Australian Gonococcal Surveillance Programme (AGSP) is a national surveillance program that has been implemented to monitor trends in *Neisseria gonorrhoeae* prevalence, incidence, and resistance [12]. The surveillance system is pivotal in the formulation of policy and action in the clinical management of *N. gonorrhoeae*. Ensuring the AGSP provides data that is actionable is critical for the development of efficacious stewardship. Despite paramount significance of the system in the management of both *N. gonorrhoeae* and antimicrobial resistance, the AGSP has been sparingly evaluated. This gap presented in systematic evaluation offers a compelling opportunity to advance and demonstrate the system’s capabilities and effectiveness.

Therefore, this paper aims to demonstrate a modified STM methodology to be used to assess the Australian *N. gonorrhoeae* surveillance as a case study to demonstrate the robustness of the methodology. The objectives are to (1) construct a scenario tree for the Australian *N. gonorrhoeae* surveillance system, (2) calculate an overall system sensitivity value, and (3) evaluate the representativeness of the system. 

## 2. Materials and Methods

This study was based on the Australian *N. gonorrhoeae* surveillance system—the Australian Gonococcal Surveillance Programme (AGSP). The AGSP collects *N. gonorrhoeae* data from a laboratory network across Australia and reports the associated antimicrobial susceptibility data and the isolate [12]. The objective of the surveillance system is to monitor the trends of antimicrobial resistance of *N. gonorrhoeae* over time to different classes of antibiotics. Further details of the surveillance system are established within surveillance reports and literature [12,13,14]. 

### 2.1. Core Methodology for Scenario Tree Building

#### 2.1.1. Modified Scenario Tree Modelling Methodology and Structure

STM is a structured stochastic approach in decision analysis to investigate the probability of a particular outcome of interest to occur given an sufficient comprehension of the preceding sequence of scenarios involved in generating the outcome of interest [10]. Previous applications of STM within the context of infectious diseases and pathogenic surveillance have primarily investigated freedom from disease [15,16,17,18]. To achieve certainty in the freedom of such disease, the paradigm championed has been to construct a series of plausible scenarios for which the presence of disease is not captured [15,16,17,18]. In modifying the STM methodology to solely focus on capturing the events in which the pathogen is detected through surveillance and excluding procedures where the disease is not captured, enables the utilization of a tool that can effectively assess the function of disease surveillance that has yet to be used.

A foundational step to building the scenario tree is to identify the key decision points and outcomes of the system [10]. These are independent events which lead to the specified desired outcome. Thus, the probability of all events occurring is given by the product of all independent events within the scenario. Relevant Australian *N. gonorrhoeae* surveillance literature was reviewed to identify the key outcomes of the scenario tree [14,19,20]. Expert and key stakeholder consultation with clinicians, in the space of sexual testing and surveillance, was conducted to identify the components and pathways of the surveillance system. To understand the intricacies of how an individual is processed by the surveillance system, topics of discussion included providers of testing, test types, and differences in testing between groups. The scenario tree was then mapped using a tree diagram representing the decision points with the relevant outcomes as per the Martin, Cameron, and Greiner [10] procedure. 

Nodes were then classified based on what they described. Definitions for each of the types of nodes can be found in Table 1. Figure 1 provides a general overview of STM.

#### 2.1.2. Collection of Data for Parameters

Following the identification of key nodes in the scenario cascade, data is to be collected for the parameterisation of the model. Targeted grey and academic literature searches were used to identify epidemiological studies to evaluate the likelihood of each decision point. In the absence of literature-based data the STM methodology allows for the use of expert opinion to obtain a likelihood estimate [10].

### 2.2. Parameterisation of Inputs for the Scenario Tree Model

Parameterisation of nodes was then completed for implementation. Parameterisation of the nodes followed the procedures outlined by Martin, Cameron, and Greiner [10]. Nodes were fit with probabilistic distributions based on their expected behaviour. Where ranges in epidemiological data were specified without a given expected value, the node was fit as a uniform distribution. For parameterising expert opinion, estimates were obtained for a minimum, mean, and maximum probability and by using a Program Evaluation and Review Technique (PERT) distribution [10]. Static values for proportions and point estimates were used elsewhere. All model inputs used with sources, descriptions, assumptions, and limitations are within the Appendix A.

### 2.3. Implementation of the Scenario Tree Model

#### 2.3.1. Core Assumptions of the Model

STM methodology has two main assumptions. The first assumption assumes perfect specificity of diagnostic testing in surveillance (a nominal probability of 1) and that there are no false positives [10]. The second assumption presumes a closed system in which all uncertain cases of *N. gonorrhoea* are resolved and a definitive diagnosis is made without the loss of individuals within the specified system to follow up or treatment as to not disrupt the sequence of specified scenarios [10]. 

#### 2.3.2. Model Outputs

The use of STM will allow for the ascertainment of the surveillance systems’ sensitivity. First, the individual components must be calculated by the equations described by Martin, Cameron, and Greiner [10]:(1)$$CSe_{j}=1-\prod_{x=1}^{x}\left(1-{SeC}_{i}\right)$$
where x represents the length of components identified in the relevant pathway for AMR detection and SeC_i_ denotes the sensitivity of the ith compartment of the scenario tree derived from parameter inputs. 

The $CSe_{j}$ values calculated can then be aggregated to provide an overall estimate of the surveillance systems representativeness. This can be evaluated in the following equation:(2)$$SSeU_{i}=1-\prod_{j=1}^{j}\left(1-{CSe}_{j}\right)$$
where j represents the number of category components in capturing AMR prevalence, $SSeU_{i}$ represents the overall sensitivity of the surveillance system branch, and CSe_j_ denotes the component surveillance at the jth stratum.

Following the calculation of surveillance system branches, aggregation of all SSeU_i_ values allows for the calculation of the overall surveillance system sensitivity. This process follows the calculation outlined in Equation (3):(3)$$SSe=\sum_{i=1}^{n}SSeU_{i}$$
where SSe is the total surveillance system sensitivity across all branches of the surveillance system.

To evaluate detection capacity for the system subcomponents, the probability of at least one pathogen detected given a set number of individual samples probability of non-detection, given by the complement of the probability of detection, over the period for which the number of individual samples were collected for surveillance was calculated with Equation (4).
(4)$$\mathrm{Pr}\left(\geq 1\mathrm{Positive}\;\mathrm{Unit}\;\mathrm{Detected}|\mathrm{Infected}\right)=1-{\left(1-CSe_{j}\right)}^{n}$$
where ${\mathrm{CSe}}_{j}$ is the sensitivity of pathogen detection for a given surveillance system component and n is the number of independent samples processed within a specified timeframe. An underlying assumption is that the processes test a representative group of independent n units. The number of required samples to be processed was derived from the 2020 Australian Gonococcal Surveillance Programme Annual Report [13]. The specified time-period was generalised to a single year using the annual proportion of isolates tested at 7222. The report stratifies the origin of isolates with associated antimicrobial susceptibility data into male and female at 5598 isolates and 1580 isolates, respectively [13]. 

#### 2.3.3. Simulation of the Model

The model was constructed in R 4.2.2 using the mc2d [21] package for statistical distributions. The model has been supplied within the Appendix A. Following construction, the model was run to capture 100,000 samples using a Monte Carlo simulation with a fixed seed. With the large sample size, the sensitivity of the *N. gonorrhoeae* AMR surveillance system could be normally approximated. Simulation parameters were specified to produce 95% confidence intervals with the mean *N. gonorrhoeae* sensitivity estimates for gonococcal and antibiotic resistance detection.

### 2.4. Modifications to Core Scenario Tree Modelling Methodology

Scenario tree modelling has yet to be demonstrated in the context of an AMR surveillance system. The core paradigm of STM has been centred in providing evidence to suggest freedom from a disease through continued surveillance. However, as *N. gonorrhoeae* and AMR are both endemic, this conflicts with the core paradigm. Consequently, key elements of the methodology must be changed to reflect the nature of the diseases.

In adapting STM to the context of AMR surveillance, a synthetic population with active *N. gonorrhoeae* infection and resistance was generated. Generated scenarios then focused solely on the events necessary for detecting and capturing *N. gonorrhoeae*. Specifically, omission of design prevalence, adjusted risk, and simulation of prolonged surveillance to prove freedom from disease [10] were replaced for epidemiological estimates of the current proportions captured by surveillance. Final calculations for system sensitivity followed the summation of the probabilities of capturing the disease as opposed to the probabilities of failure. The proposed paradigm in which STM is to be applied to the *N. gonorrhoeae* AMR surveillance system focusses on the probability for which an individual carrying a resistant organism is detected through the defined sequence of the model. In the interpretation of the system sensitivity values output by Equations (1) and (2), scenario tree modelling will produce an expected proportion of infected individuals captured by the surveillance system.

### 2.5. Sensitivity Analysis of the Scenario Tree Model

The sensitivity analysis process followed two stages. The first stage was parameter importance, whereby sensitivity analysis was completed in ModelRisk [22] using the sensitivity analysis function with a reimplemented model to determine the influence of variables. The model has been supplied in the Appendix A. The second stage examined the effect on overall system sensitivity by manually altering variables in permutations to simulate a systematic change (i.e., policy and intervention) and facilitate the identification of leverage points. For this study, thematic areas in the model will be identified based on common characteristics and have probabilities altered at values of 0.1, 0.5, and 0.99. To further identify key leverage points, ModelRisk’s [22] sensitivity analysis tool will allow for the determination of individual parameter influence on the model output.

## 3. Results

### 3.1. Scenario Tree Model for the Surveillance System

The scenario tree maps in Figure 2 and Figure 3 illustrate the pathways for diagnosing *N. gonorrhoeae* and determining its antibiotic resistance status. Using the STM methodology and procedures the AGSP was found to encompass 16 distinct dichotomous branches for detecting gonococcal infection and antibiotic resistance. These 16 pathways represent the subcomponents of the surveillance system.

There were four distinct category nodes that encompassed thematically grouped branches by the intricacies of *N. gonorrhoeae* infection. The category identified the stratification of infected individuals by biological sex and clinical symptom status. This grouping considers the differential risk factors of sex and whether an individual is symptomatic. The second category is based on the differential risk of health-seeking behaviours of infected individuals. The dividing factor of health-seeking behaviour was contingent on an individual’s sex. The third category identifies the differential risk of initiating clinical testing. The dependencies of this node were biological sex of the individual and the respective healthcare setting. The final division identified convergence of all pathways through diagnostic laboratory testing to confirm the presence of *N. gonorrhoeae* and determine antibiotic resistance with antibiotic susceptibility data.

The results of the STM indicated that for health-seeking behaviours there were four main locations for STI testing as indicated by the literature [23] and expert opinion. This includes general practice (GP) clinics, sexual health clinics, community health clinics, and testing in tertiary care. Physiological differences by sex were determined to influence the individual interpretation of symptoms. Because of the differences in symptom presentation, the probability inputs for health-seeking behaviours would be influenced. This resulted in two separate pathways for males and females.

The diagnostics embedded within laboratory components of the scenario tree are presented within Figure 3. No significant differences were substantiated within the given parameters by biological sex. The processes described highlighted a dependency on clinician-initiated testing for the ascertainment of *N. gonorrhoea* and antibiotic resistance status. In determining antibiotic resistance status and obtaining epidemiological reports, the sequence of events evident within the scenario tree indicated a dependency on antibiotic susceptibility testing data being present.

### 3.2. Scenario Tree Model Outputs from the Australian Gonococcal Surveillance Programme

#### 3.2.1. Overall System Sensitivity Outputs

The STM simulation results for the AGSP in detecting *N. gonorrhoeae* and antibiotic resistance can be seen in Table 2. The complete model in R can be examined in the Appendix A. The distributions for the simulations with 95% confidence intervals are depicted in Figure 4. Sensitivity explicitly refers to the expected proportion of infected individuals captured by the surveillance system. The parameter space for sensitivity is between 0 and 1, denoting the percentage infected individuals detected. The results of the STM in the surveillance system has demonstrated the AGSP was sensitive to the detection *N. gonorrhoeae* at an expected sensitivity of 0.624 (95% CI; 0.524, 0.736). The sensitivity value was greater than the mean sensitivity of the antibiotic resistance surveillance system, which was 0.144 (95% CI; 0.106, 0.189). For gonococcal detection, the system was observed to produce the greatest maximum of 0.848. The lowest sensitivity observed was the minimum of antibiotic resistance at 0.08. The range of values observed in the detection of *N. gonorrhoeae* imply the system is expected to capture approximately 62.4% of the infected population, with the confidence interval also supporting the potential for detection to be between 52.4% and 73.6% of the true population. Furthermore, in the context of ascertaining antibiotic resistance, the model anticipates 14.4% of the infected individuals to have their susceptibility status identified and reported to the surveillance system. The confidence interval produced supports the range to be between 10.6% and 18.9%. The two distributions differ by the inclusion of the antibiotic resistance detection parameters, antimicrobial susceptibility (AST) data, and sensitivity of culture. The effect of the parameters caused the narrowing of the distributions resulting in a smaller confidence interval. 

#### 3.2.2. System Sub-Component Sensitivity Outputs

The system component sensitivities are the sensitivities of the individual subcomponents that constitute the overall system. Subcomponents are the branches of the scenario tree model. The system component sensitivities for gonococcal detection and antibiotic resistance are presented in Figure 5. Raw outputs can be viewed in the Appendix A. 

The greatest surveillance system component sensitivity was observed within symptomatic males attending general practices. The lowest observed component sensitivity was within asymptomatic females presenting to tertiary care. Overall trends across *N. gonorrhoeae* infection and resistance indicate across all surveillance system components, with the exception of tertiary care settings, the surveillance system consistently detects a greater proportion of males as compared to females. This pronounced disparity in gender representation within the system subcomponents raises concerns regarding the potential for sex-based reporting bias to be present within the system. In observing the presence of symptoms, the model indicates a greater proportion of individuals captured would be symptomatic in nature as compared to asymptomatic. Sexual health clinics as a component of surveillance have been shown to exhibit high sensitivity to asymptomatic populations as compared to other settings which positions them as an optimal healthcare setting for asymptomatic detection. Other notable trends highlight that general practice yields the highest system subcomponent sensitivity across biological sex. Asymptomatic females had a higher probability of being detected than asymptomatic males for both gonococcal infection and antibiotic resistance.

### 3.3. Sensitivity Analysis

#### 3.3.1. ModelRisk Sensitivity Analysis

The full sensitivity analyses as produced by ModelRisk (VoseSoftware version 6.1.94) can be viewed in the Appendix A. The sensitivity analysis depicting the influence of each parameter on the sensitivity value in the overall model can be seen in Figure 6. Among the evaluated parameters, the presence of AST data as a model parameter has emerged as the most influential factor in ascertaining antibiotic resistance status. Increasing the variable to the maximum specified value had the most pronounced effect on system sensitivity. This was followed by symptomatic male healthcare access and laboratory diagnostics such as culture and diagnostic test sensitivity. Parameters concerning males were observed to exert a greater influence on the overall system sensitivity as compared to parameters concerning females. As a generalisable trend, parameters pertaining to females exerted a near negligible effect on the overall system sensitivity for antibiotic resistance.

#### 3.3.2. Modification of Parameters Identified from Sensitivity Analysis

ModelRisk’s sensitivity analysis identified AST, health service access, and clinician testing as influential parameters for modification due to their influence of surveillance system sensitivity when modified. Figure 7 illustrates the result of modifying identified parameters at 0.1, 0.5, and 0.99 in various permutations to simulate both detrimental and supplemental activity to increase influence. Appendix A presents the raw results of input parameter modification at values of 0.1, 0.5, and 0.99 for overall system sensitivity. Notably, the greatest increase in overall system sensitivity was observed when the parameters of AST, health service access for symptomatic and asymptomatic patients, and clinician testing were set to 0.99. The findings suggest that, given all asymptomatic and symptomatic patients were screened with AST testing completed, it would be expected that 85% of the infected population would have their resistance status ascertained. Conversely, modification of parameters at values of 0.1 significantly lowered overall system sensitivity. Computing system sensitivity with values of the AST parameter set at 0.5, and 0.99 was observed to result in the greatest increase caused by a single parameter. An increase in system sensitivity for antibiotic resistance detection was observed when modifying AST data in any permutation from the values of 0.5 and above. Non-significant impacts on overall system sensitivity were observed in joint modifications of health service access and clinician testing.

### 3.4. Estimation of Detection Capability

The detection capacity measured by the number of samples required for absolute certainty of detecting an exotic antibiotic resistant serotype is presented in Figure 8 for each system component. It was observed that the lowest number of samples needed to detect an exotic serotype in the symptomatic populations in general practice settings was observed in symptomatic males who attended general practice clinics, with an estimated 49 isolates required for exotic serotype detection. Given the epidemiological reports notifications for *N. gonorrhoeae* in Australia [24], the model would strongly support the notation that the initial detection of an exotic serotype would be detected from a general practice clinic, assuming comprehensive serotyping. For the asymptomatic population, the values produced far exceed the notifications currently presented within *N. gonohorreae* epidemiological reports [24], suggesting an implausibility for an exotic serotype to first be detected within a tertiary healthcare context. The greatest number of units required was observed in the tertiary care setting for both males and females. The results suggest that the number of samples required for testing varies inversely with symptomatic status. Specifically, fewer samples are required to be drawn from males with symptoms as compared to females, while for asymptomatic status the number of samples needed to be drawn from females is lower as compared to males.

Figure 9 displays the annualised probability of each surveillance system sub-component to detect an exotic, resistant *N. gonorrhoeae* serotype over a single year. The results demonstrate that the highest and most consistent annualised probabilities of detecting an exotic serotype are observed within the symptomatic male population, regardless of healthcare service. Conversely, the symptomatic female population displays uniformly lower probabilities across surveillance system components. Notable findings include the lowest annualised probability, which was found in asymptomatic males receiving tertiary care. Sexual health clinics emerged as a reliable and consistent healthcare setting for detecting *N. gonorrhoeae* infections for both asymptomatic and symptomatic populations as indicated by the relatively low number of isolates required in both contexts. It is to be acknowledged that this implication is contingent on timeliness being defined over an annualised period.

## 4. Discussion

The improvement of surveillance for *N. gonorrhoeae* and its resistance is imperative to furthering the efficacy of disease control endeavours. At the time of publication, this study showcases a novel application of STM to systematically deconstruct and evaluate an AMR surveillance system. The modelling process has highlighted aspects of surveillance data generation that have not been considered previously through the inclusion of differential risk due to sex, health-seeking behaviour, and clinician behaviour as critical steps within the process. Interestingly, this subverts the conventional AGSP structural conceptualisation as a solely laboratory-based network [12] by identifying the antecedent events for a potential sample to be generated. Indeed, this result would suggest the existence of further complexity that has not previously been acknowledged but is pertinent to how samples are generated. Utilisation of this information may encompass the construction of auxiliary systems to better understand the intricacies of sample generation for greater confidence in reported surveillance figures.

The system sensitivity outputs of the model indicate an underestimation of prevalence for both gonococcal and antibiotic resistance detection. The finding of underestimation presents insignificant novelty in the discussion of *N. gonorrhoeae* surveillance [25]. However, this study presents an objective quantification of underestimation. This study has demonstrated that, in a simulated population of infected individuals, there exists considerable difference in the proportion of captured *N. gonorrhoeae* infections and resistant isolates. A structural criticism to be argued from the system sensitivity outputs suggest the inappropriateness of a clinical system for the detection of antibiotic resistance. The substantially greater system sensitivity of gonococcal detection through the exclusion of AST data as a parameter emphasises the unsuitability of structuring antibiotic resistance ascertainment as an optional additional step on an ad hoc testing regime. Potential changes in implementing routine testing may enhance surveillance by increasing the number of isolates with associated AST data, and thus increasing sensitivity. For instance, influenza-like illness surveillance carried out by the Australian Sentinel Practice Research Network (ASPREN) has implemented a standardised protocol for collecting swab samples from presentations on a designated day [26]. Indeed, this modification is feasible but requires further examination of laboratory capacities to implement such changes.

An interesting result revealed by the system sub-component sensitivities is the presence of biased sampling. The observed trend of higher sensitivities in diagnostic components related to males, as compared to females, and in symptomatic individuals, as compared to asymptomatic individuals, can be interpreted as greater proportions being detected through these pathways. This would indicate that sampling for isolates would inherently present greater probability for the sample to originate from a symptomatic male. For detection of *N. gonorrhoeae* infection, this result is in line with broader epidemiological literature suggesting greater prevalence amongst men [27,28]. However, there are concessions to be made regarding the probability of resistance susceptibility. In the discussion of antibiotic resistance, there is a growing body of literature which may suggest gender influences an individual’s susceptibility to resistance [29,30,31]. Thus, there is an argument to acknowledge the bias presented. Enhancement of the epidemiological understanding can be conducted by utilising supplementary systems to the surveillance structure such as surveys. However, there is a clear need for additional research to delineate pragmatic benefits in implementing these systems.

The sensitivity analyses provide potential leverage points for enhancement of the surveillance system. The parameter for AST data was found to be the most influential in the model’s sensitivity outputs. Hence, identifying interventions to increase the number of samples with associated AST will substantiate greater sensitivity of the system. Moreover, as indicated by the manual sensitivity analysis, the modification of clinician behaviour and AST data yielded substantial increases in overall system sensitivity for antibiotic resistance detection. Both steps are related in their clinician-initiated nature. From a pragmatic perspective, a viable strategy is to focus on effecting systematic change amongst clinicians. This population group is most feasible, as clinical guidelines represent a readily available tool to influence behaviours [32]. The results of guideline modification may substantiate a greater proportion of samples having a parallel culture initiated for resistance status. Overall, this would result in greater system sensitivity and improve confidence in figures reported by the surveillance system.

The detection capability of the system to capture exotic serotypes over an annualised period has been demonstrated to be feasible, given sufficient isolates are provided. In the discussion of the surveillance system’s detection capability, a pertinent topic is the detection of exotic multi-drug resistant *N. gonorrhoeae.* A relevant example has been documented by Whiley, et al. [33] whereby the simultaneous isolation of an exotic ceftriaxone and azithromycin resistant gonorrhoeae strains in symptomatic patients has been documented. The model demonstrates that, if surveillance prioritises detection, given the volume of notifications remains constant as presently reported, there exists substantial confidence in detecting exotic serotypes. This finding does not consider the aspect of timeliness required when adjusting clinical management guidelines if an exotic species is detected. Additional evaluation of laboratory sample processing capabilities would be required to substantiate the finding for detection capacity.

### Strengths and Limitations

The primary strength of this study is encompassed by the novelty of the assessment methodology. This study has employed an objective stochastic modelling approach to evaluate the surveillance system. The integration of diverse data sources facilitates a deeper understanding of the intricacy involved in data generation. The approach demands a more comprehensive examination of surveillance beyond the superficiality of the system. Furthermore, the model inputs and outputs are easily readable. Leverage points can easily be identified through the sensitivity analysis. Model inputs can then be modified to imitate the effects of public health policy and inform future strategies. From there, evaluation of surveillance strategies could be further extended to include cost-effectiveness which has been demonstrated previously in the relevant literature.

There are limitations to consider in the model and design of this study. The model is not exhaustive and lacks stratified parameterization for health-seeking and clinician behaviours based on social and economic determinants implicated in *N. gonorrhoeae* infections [34]. This resulted in an incomplete characterization of *N. gonorrhoeae* diagnosis and limits the specificity of the findings. Future research should aim to incorporate more comprehensive and stratified parameterization based on further epidemiological data to improve the accuracy and explanatory power of models. Despite this limitation, the model has accommodated for uncertainty within the parameters to support its generalisability. Moreover, identified leverage points within the sensitivity analyses provide a linearised view of the parameters. In the presented application of STM, it is to be acknowledged that the assumptions of perfect specificity are idealistic and will influence interpretability of the sensitivity outputs produced. Estimates for surveillance sensitivity may be greater than in real contexts. To further mitigate the influence, adjustments for specificity, based on data, will be required. In appreciation of the complexity, leverage points identified are at a macroscopic level. The model has an inability to properly represent the dynamic relationships between variables that may exist. As such, the model only identifies general areas for improvement. For targeted action, approaches which can incorporate the dynamic relationships would be more appropriate.

## 5. Conclusions

This study presents a novel application of scenario tree modelling to evaluate antibiotic resistance surveillance systems. The methodology is robust and can be readily adapted to related disease contexts for antibiotic resistance. Overall, the evaluation of the AGSP using the STM methodology has yielded mixed findings. This study indicated an adequate detection capability of the surveillance system over an annualised period to capture an exotic *N. gonorrhoeae* serotype given the current notification. The model has substantiated evidence that may challenge the appropriateness of clinical testing systems in effectively monitoring antibiotic resistance. Coupled with the identification of systematic bias, the work presented has wider implications concerning the representativeness and subsequent interpretation of surveillance reports. The estimation of antibiotic resistance prevalence may be substantially underestimated without further examination. This study provides evidence to advocate for the implementation of a systematic testing routine for *N. gonorrhoeae* samples as a strategy to enhance surveillance sensitivity and improve the representativeness of the system. Alternatively, this study highlights targeting clinician behaviour to increase the proportion of samples that have initiated culture as another strategy to improve sensitivity. Further research in improving the specificity of the model may elucidate greater insights to the findings presented within this study.

## Figures and Tables

**Figure 1 pathogens-12-00907-f001:**
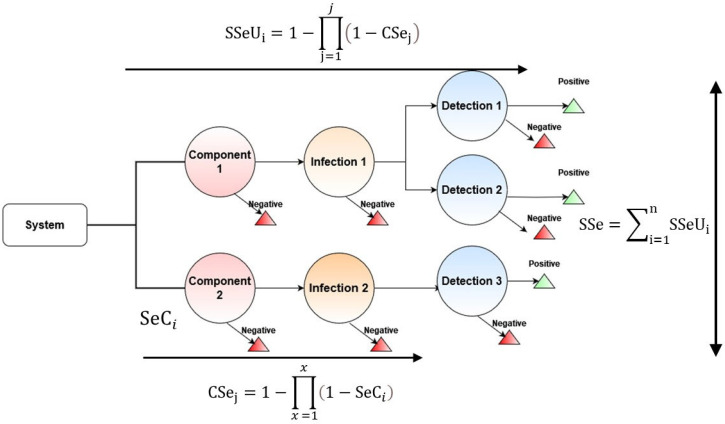
General overview of the scenario tree model (STM).

**Figure 2 pathogens-12-00907-f002:**
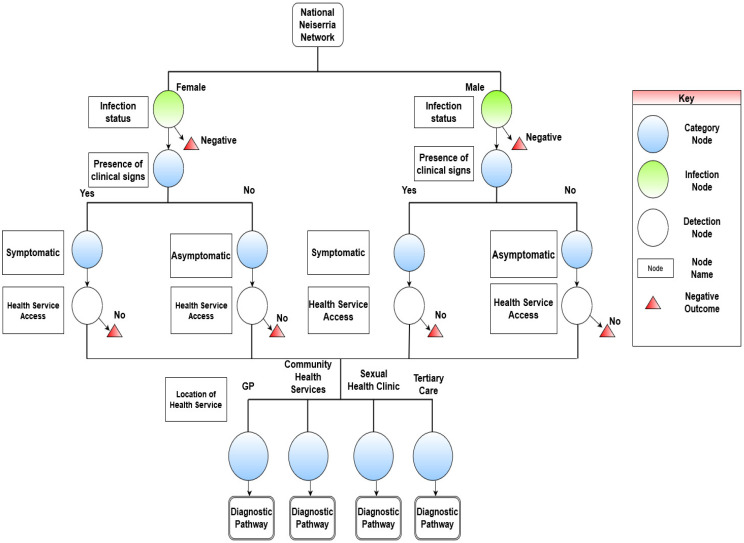
Scenario tree illustrating the health-seeking behaviours required to enter the Australian Gonococcal Surveillance Programme (AGSP) to capture *N. gonorrhoea* and antibiotic resistance in Australia.

**Figure 3 pathogens-12-00907-f003:**
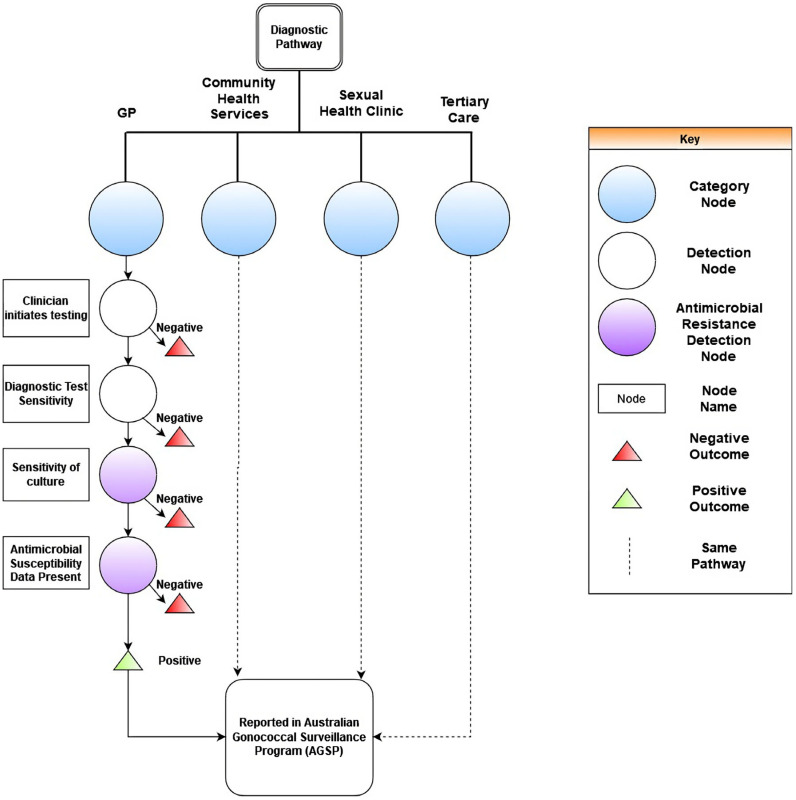
Continuation of scenario tree for the Australian Gonococcal Surveillance Programme (AGSP) to capture *N. gonorrhoeae.* Depicted in the figure are the parameters for clinician behaviours and laboratory testing.

**Figure 4 pathogens-12-00907-f004:**
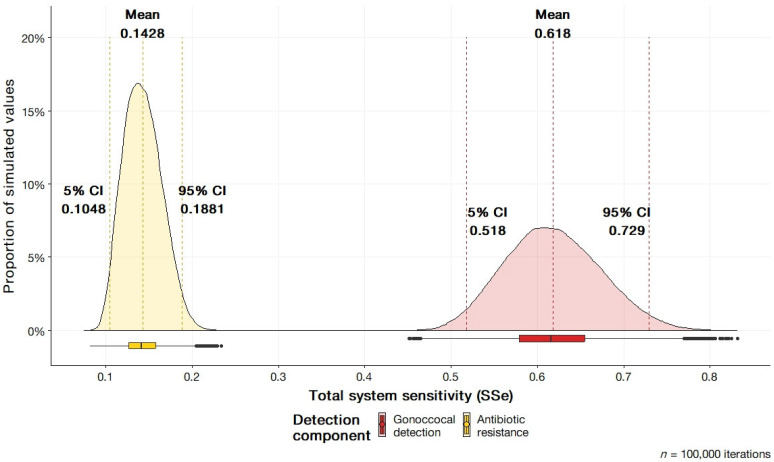
Total system sensitivity (SSe) for gonococcal and antibiotic resistance detection components of the Australian Gonococcal Surveillance Programme (AGSP).

**Figure 5 pathogens-12-00907-f005:**
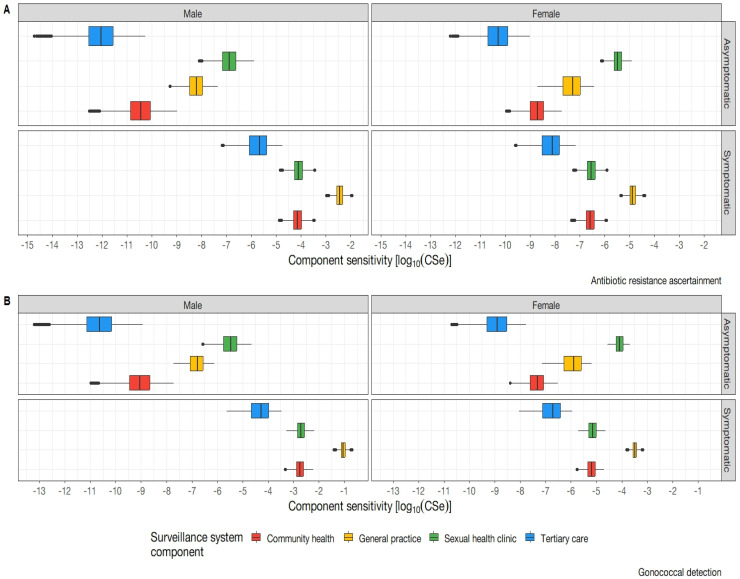
Sub−system component sensitivity (CSe) on a logarithmic scale. (**A**) depicts the component sensitivities by sex, symptom status, and health service for the determination of antibiotic resistance. (**B**) Depicts the component sensitivities for gonococcal detection by sex, symptom status, and health service.

**Figure 6 pathogens-12-00907-f006:**
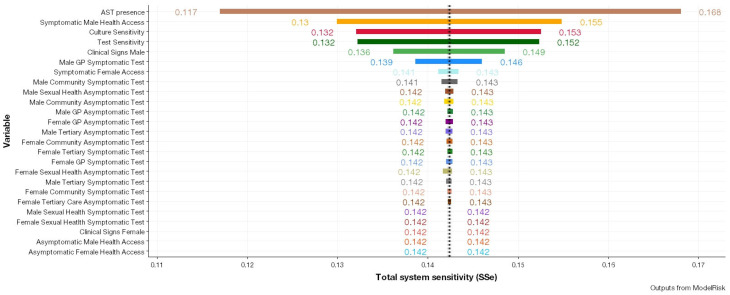
Results of the sensitivity analysis through modification of different parameters represented by different colours pertaining to antibiotic resistance detection and the effect on overall system sensitivity (SSe).

**Figure 7 pathogens-12-00907-f007:**
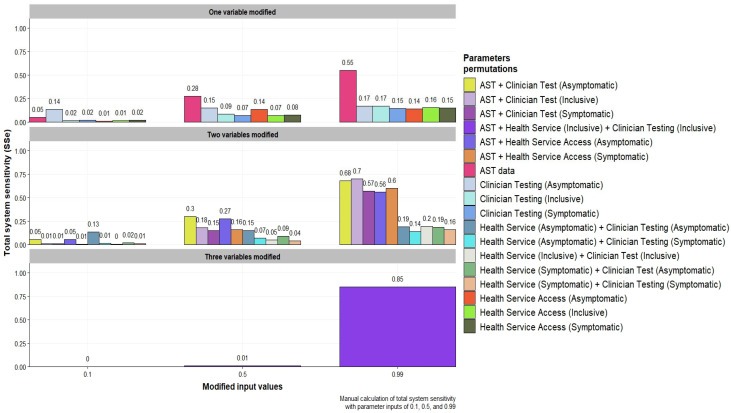
Total system sensitivity (SSe) for antibiotic resistance detection computed with probabilities values of 0.1, 0.5, and 0.99 into identified surveillance components $\left(SeC_{i}\right)$ in varying permutations within the scenario tree model.

**Figure 8 pathogens-12-00907-f008:**
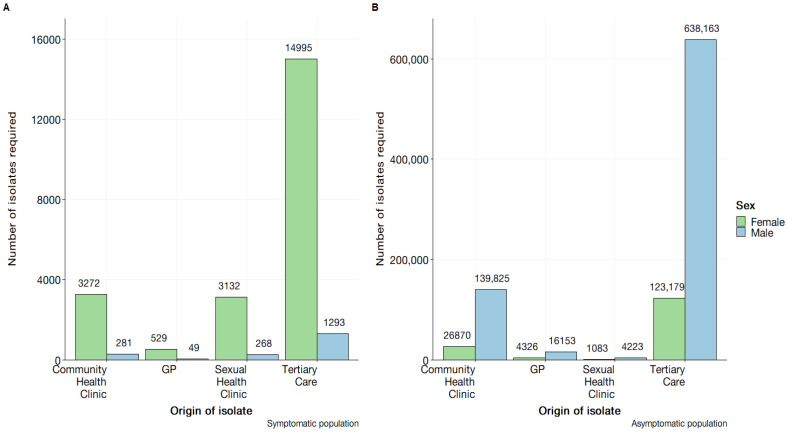
Demonstration of the estimated number of isolates required for antibiotic susceptibility testing for absolute certainty in exotic serotype detection using component sensitivity (CSe) and calculated by $1-{\left(1-CSe_{i}\right)}^{n}$. (**A**) Contains the symptomatic male and female component sensitivities by healthcare setting. (**B**) Contains the asymptomatic male and female component sensitivities by healthcare setting.

**Figure 9 pathogens-12-00907-f009:**
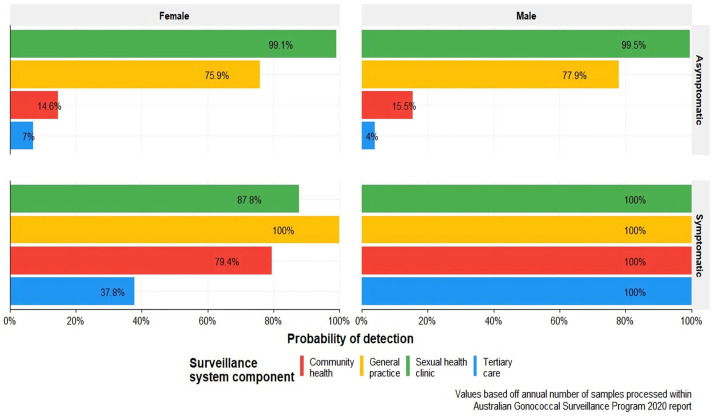
Annualised probability of surveillance system components detecting an exotic serotype based on case load presented in the 2020 Australian Gonococcal Surveillance Programme report [13].

**Table 1 pathogens-12-00907-t001:** Scenario tree modelling node classifications with their respective definitions.

Node	Definition
Antimicrobial Resistance Detection Node	Refers to the points at which *N. gonorrhoea* antibiotic resistance is detected. Given as a dichotomous event.
Category Node	Category nodes refer to proportions of a population that fall on a given pathway.
Detection Node	Detection nodes are points at which *N. gonorrhoea* is detected. Given as a dichotomous event.
Infection Node	Infection nodes refer to the reported proportion of infections for the specified group.

**Table 2 pathogens-12-00907-t002:** Summary output statistics for the scenario tree model.

System Component	Minimum	Lower 95% CI	Mean	Median	Upper 95% CI	Maximum
Gonococcal detection	0.457	0.524	0.624	0.625	0.735	0.848
Antibiotic resistance status	0.08	0.106	0.144	0.143	0.189	0.23

## Data Availability

All data are presented within this study’s Appendix A.

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
