# Peer review of "Enhancing Insights into Australia’s Gonococcal Surveillance Programme through Stochastic Modelling"

_pathogens, 2023, doi:10.3390/pathogens12070907_

Round 1
Reviewer 1 Report
The article presents an adaptation of an existing method - Scenario Tree Modelling (STM) - applied to a novel context (Gonococcal infection incidence, including anti-microbial resistance, in Australia). This includes extending the STM approach from the context of population-free-from-disease, to the situation of an endemic disease.
The article is well structured and at a high level is very clear about what has been done, what the findings are, and what the possible implications are. It was a pleasure to see a clearly articulated (and addressed) research question.
At a more detailed level, the article could benefit from including a little more explanatory, methodological detail about STM as a general approach and how it is used (and adapted) specifically in this context. The article cited as [10] in this MS is an excellent source of material to start from for trying to pull together such an explanation.
Some of the figures in the article could be improved. Figs 1 & 2 could benefit from more labels being a little larger; or alternatively for the figures being larger themselves - they are informative enough to warrant each figure being sized to fill a page. As a side note, the text of the article, and the reference [10] discuss STMs as represented by dichotomous decision trees. Although the infected/not and AMR detected/not nodes are dichotomous, others like the category nodes are clearly not dichotomous. I.e. the nodes that correspond to a proportions rather than e.g. detected/not probabilities.
Some minor notes:
- Some of the notation is inconsistent (or if it is not, then it is difficult to interpret correctly) e.g. SeC and CSe are both used - I am assuming that both are intended to indicate sensitivity for/at the component level. (More generally, the variables used in equations could be better defined to ensure that they are unambiguous in all cases). It also looks like there are instances where indices swap between i and l when they are they same thing.
- The authors note some fundamental assumptions of the STM approach. As noted by the reference [10] these assumptions won’t always hold. How well do these assumptions hold in the application presented here? What are the implications for the analysis if they don’t hold?
- It may simply be an issue of font, but the abbreviation used for the Confidence Intervals in the results tables looks like CL rather than CI. (I’m not sure how well the font used in the journal review portal will represent the I vs L - apologies if this is unclear.)
- The results in figure 4 are presented on different scales which makes it difficult to compare between the two types of detection being compared. Additionally, the width of the markers for several of the data series means that it is difficult to determine what colour and hence what data series are where.
- Figure 6 holds valuable information but is very difficult to gain any insight from. The large number of colours makes it virtually impossible to match series to the key explaining what they are. Numeric labels are overlapping (hence difficult to read) and the data is not explained in the figure caption (or the body text). More generally, figure captions in the MS could be more informative.
- Figure 7 also presents interesting results, in a form that makes them less clear than they might be. Presumably, the shape of the curves in all facets of Fig 7 are identical, up to a horizontal rescaling (if this is not the case, then the figure is not managing to reveal any differences). In this case the useful information captured in the facets is the different values for the number of isolates required for detection to be certain. But because each facet has a different horizontal a scale, it is not possible to visually compare between facets - which would seem to be the point of the figure. One possible alternative could be a bar chart, or a table presenting the “isolates required fro certain detection” values. A log scale may be necessary because of the rage of values. (Or a table of values may be sufficient). If the shape of the curve matters, and if it is invariant after rescaling, then a single curve would suffice.
- Some of the results could be accompanied by a brief explanation or discussion of what they imply. Although the Discussion section of the article is good it doesn’t (and probably shouldn’t) capture the implications of the all of the results that are highlighted in the preceding section.
It is great to read that the authors have made the code and models used in this work freely available. The short time window for peer review means that I haven’t had the opportunity to test and check that this works as described or the quality of the documentation, particularly whether it is sufficient for new researchers to adapt and apply the same approach for different applications.

The writing is largely good, however there are a few places where some minor edits would significantly improve readability. Specifically:
- There are a number of instances where words are either missing or where there are superfluous words.
- There are a few places where there is a mismatch between a singular/plural subject or object and the grammar of the rest of the sentence.
- There are some places where clauses in sentences are split, or arranged in such a way that readability is decreased.
- Some software (R and the mc2d package) is not cited; other software (ModelRisk) is cited in a different format to the rest of the citations in text (i.e. author names in text, rather than number-in-brackets).
Additionally, clarity of the figures could be improved by taking care that text in them is large enough to be easily read - i.e. at a size comparable to the body text of the article.
I will attach a marked up PDF that indicates the edits that I noticed, but there are likely further, similar edits. A close read of the text for grammatical edits would be desirable, before publication.
Author Response
Please see attached. We thank you again for taking the time out to review our manuscript. We have taken aboard the changes and advice you have input. We believe that it has vastly improved the quality of the manuscript.
Thank you again from the authors. We appreciate your effort.

Reviewer 2 Report
In this paper the authors aim to use the Scenario Tree Modelling (STM) approach to assess the Australian Gonococcal Surveillance Programme’s representativeness as a case study.
The manuscript report interesting results.
Major comments:
1 - Although explained in reference 10, a refresh of the method is necessary to understand. The method presented in section 2 MM needs to be expanded to better explain the method and clearly highlight the changes made to the original method.
2 - Conceptually, the method presented in Ref. 10 was designed to demonstrate freedom from disease, what modifications were made to adapt the approach to the surveillance problem? From this point of view, section 2.4 must be more precise in relation to the method elements presented further before.
3 - All the illustrations (Figs. 1 to 8) of section 3 results are not legible which makes understanding difficult
Specific comments:
4 - Add a list of abbreviations
5 - line 98, Table 1: what is "Title 3"
6 - In addition to Table 1, add a figure of a scenario tree to accompanying the description of the method.
7 - Lines 117 to 119: On the assumption of "no false positives", how can this assumption be relaxed and what would be the impact of that ?
8 - Lines 119 to 121: The second assumption is not clear
9 - All the section 2.3.2 Model outputs: Not need to understandable
10 - Not understandable, this section requires more explanation with the contribution of a figure
11 - Lines 451 to 453: revise the "Acknowledgements" of disregard it
Author Response
Please see attached. We thank you so much for helping to improve the manuscript. We greatly appreciate you taking the time and effort out to read our work. We believe with your inputs, this has greatly contributed to our manuscript's readability.

Round 2
Reviewer 2 Report
The authors have satisfactorily responded to all my comments. No further questions from my side.